# Spatial and Temporal Agglomeration Characteristics and Coupling Relationship of Urban Built-Up Land and Economic Hinterland—A Case Study of the Lower Yellow River, China

**Yunfeng Cen** [1,2], **Pengyan Zhang** [1,2,3,*], **Yuhang Yan** [1,2], **Wenlong Jing** [4,5,6,*] 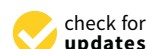, **Yu Zhang** [1,2], **Yanyan Li** [1,2], **Dan Yang** [1,2], **Xin Liu** [1,2], **Wenliang Geng** [1,2] and **Tianqi Rong** [1,2]

[1] Key Laboratory of Geospatial Technology for Middle and Lower Yellow River Region, Research Center of Regional Development and Planning, Henan University, Kaifeng 475004, China; 18203999416@163.com (Y.C.); 15736873184@163.com (Y.Y.); 17839221663@163.com (Y.Z.); yyli1109@163.com (Y.L.); Yangdan219@126.com (D.Y.); qhdliuxin@126.com (X.L.); gengwenliang418@126.com (W.G.); rong122030@163.com (T.R.)

[2] Institute of Agriculture and Rural Sustainable Development, Henan University, Kaifeng 475004, China

[3] Collaborative Innovation Center on Yellow River Civilization of Henan Province, Henan University, Kaifeng 475001, China

[4] Guangzhou Institute of Geography, Guangzhou 510070, China

[5] Key Laboratory of Guangdong for Utilization of Remote Sensing and Geographical Information System, Guangzhou 510070, China

[6] Guangdong Open Laboratory of Geospatial Information Technology and Application, Guangzhou 510070, China

[*] Correspondence: pengyanzh@126.com (P.Z.); jingwl@lreis.ac.cn (W.J.)

**Abstract:** Clarifying the development relationship between urban built-up land and economic hinterland can provide decision support for regional sustainable development. Using the improved field model, geographic concentration and elasticity coefficient, and taking the lower reaches of the Yellow River as the study area, this study defines the scope of urban economic hinterland in the lower reaches of the Yellow River in China from 2005 to 2017, further analyzes the dynamic process of urban built-up land agglomeration and economic hinterland agglomeration, and reveals the spatial–temporal coupling relationships of urban built-up land expansion and economic hinterland development. The results revealed the following: (1) From 2005 to 2017, the spatial and temporal change pattern of the economic hinterland of each city in the lower reaches of the Yellow River had basically the same trend, showing a low degree of coincidence with the administrative divisions and obvious differences in scope and change; (2) The geographic concentration of built-up land showed a trend of centering on the cities of Zhengzhou and Jinan, spreading to the periphery, and gradually forming a high-value contiguous area in both cities. The spatial distribution patterns of the geographic concentration of economic hinterland are mainly manifested in Zhengzhou and Jinan, showing a circle structure of areas with "highest-low-higher" concentrations of economic hinterland moving away from the urban center; (3) The spatial–temporal coupling between the expansion of urban built-up land and the development of economic hinterland underwent an obvious transformation process. From 2005 to 2011, the coupling mode was mainly growth, and, from 2011 to 2017, it began to shift to extensive and intensive development. The coupling model of urban built-up land and economic hinterland in the lower reaches of the Yellow River has a good trend.

**Keywords:** economic hinterland; built-up land; agglomeration characteristics; coupling relationship; the Lower Yellow River

## 1. Introduction

With the continuous advancement of global urbanization and regional integration and the rapid development of the economy and society, the breadth and depth of interaction among cities are increasing, which leads to the gradual transformation of cities from decentralized growth poles to more advanced urban agglomeration forms [1,2]. Urban development is closely related to a city's surrounding areas; various economic factors gather in space by the city as a node [3], while relying on the city to radiate its own influence on the periphery [4]. The hinterland is a spatially continuous area [5]. The regional scope in which the economic attraction and radiation of a city play a dominant role in the economic activities of the surrounding areas is called the economic hinterland of the city, and is also known as the radiation zone of the city, the spheres of influence, the circle of influence, etc. [6–8]. Accurate identification of urban economic hinterland and accurate identification of temporal and spatial morphological characteristics are not only conducive to grasping the spatial and temporal development direction of the city's own economy, but also help to provide data for the formulation of urban sustainable development strategies, and at the same time promote regional economic coordination. Therefore, the scope of urban economic hinterland has been a key research content of urban geography and regional economics [9].

Western countries studied the urban hinterland relatively early, from the concept and theory of central place [10] to the application of the retail gravitation model [11] and the method of actual investigation and verification [12], then to the application of the field model in the division of urban impact areas [13,14], the Voronoi diagram [15], the Hoshen–Kopelman algorithm [16] in the detection of the spatial range of urban agglomerations, and so on. After the 1990s, economic globalization and the science and technology revolution in information technology exerted great pressure on urban influential hinterland research, which tended towards networks and compartmentalization [17], such as Zeev's research on the city of Be'er Sheva in Israel from a domestic and global hinterland perspective to analyze the impact of urban sustainable development [18]. Tiago and others, dedicated to the case of container terminals in Portuguese ports, showed the main hinterlands of terminals, the impact of intermodal terminals in developing port regionalization and the contestability levels across the hinterland [19]. Nowadays, most Western countries have entered a post-industrial phase, with a highly developed economy and society. The "node-hinterland" pattern has been replaced by a "node-node" pattern [20]. So, it is undeniable that research on urban influential hinterland is diminishing in western countries [17]. However, developing countries are still in the process of industrialization, and the "node-hinterland" pattern is still dominant [20,21]. China is in a period of rapid urbanization, and urban agglomerations have become the focus of regional economic and urban polarization [22]. Therefore, Chinese scholars began their research on urban hinterlands relatively late, which has only started to emerge in recent decades. In terms of research methods, most of them used the field strength model [23,24], breakpoint model [25,26], Huff model [27,28], and Bass model [29]. In terms of the level of regionalization, the research areas have involved nations [23], provinces [30], Economic Zones [31,32], and Urban Agglomerations [33,34], etc. The fruitful results of these studies have made great contributions toward enriching the theory and practice of urban economic hinterland in China.

However, in terms of research content, most scholars have focused only on empirical research to define the scope of urban economic hinterland, without exploring the relationships between urban economic hinterland and other development factors, such as the growth of urban built-up land. Although Li [35] and others concluded that the introduction of urban land can expand the scope of high urban economic hinterland, they still could not explain the spatial and temporal development relationship between urban economic hinterland and urban land use. In developing countries such as China, there is a strong correlation between urban economic hinterland and built-up land [36]. On the one hand, driven by technological progress, the economic hinterland expands to a broader area, which makes the circulation of goods, people and capital between the urban center and the economic hinterland more frequent. As a result, built-up land such as residential, industrial and commercial land, and transportation infrastructure and so on spread to the periphery of the city. This can be

explained by the fact that the economic hinterland is the external driving factor of the expansion of built-up land [1,37]. On the other hand, while the built-up land expands to the periphery of the city, it often forms a close connection with the city center, which provides more factors of production for the city, thus affecting the spatial pattern of the urban economic hinterland. This can be explained by the fact that the expansion of urban built-up land is also one of the leading forces of economic hinterland growth in a certain stage [38]. Therefore, it is necessary to explore the spatial–temporal development relationship between urban built-up land and economic hinterland.

The lower reaches of the Yellow River, located in the central and eastern regions of China, are one of the typical areas where China's economic development is relatively rapid and human activities have a more significant impact on land use [39]. In recent years, cities in the lower reaches of the Yellow River, especially the big cities, have been highly focused on their economy and show multi-center expansion in their spatial form, which has extended the hinterland of urban economy to the surrounding cities, creating a pattern of a closer economic network, with less obvious boundaries and boundaries between cities. At the same time, with the economic development and urban expansion, urban built-up land is also undergoing more dramatic changes, especially in Zhengzhou and Jinan—the two major provincial capitals, which are headed by "blowout" development. Economic growth and the expansion of built-up land have led to a more prominent contradiction of land use in the lower reaches of the Yellow River, which is also one of the serious problems facing urban development in China [37], and has a high similarity with the urban development of some developing countries and regions in the world [40]. Therefore, based on the regional spatial unit of the lower reaches of the Yellow River, using ArcGIS 10.2 software as the technical platform, with the help of a field strength model, geographic concentration model and elastic coefficient model, this paper further analyses the urban built-up land and economic hinterland collection on the basis of defining the economic hinterland of the lower reaches of the Yellow River from 2005 to 2017. The dynamic change process of agglomeration reveals the space-time coupling relationship between urban built-up land expansion and economic hinterland development. The research results reflect the coordination between urban built-up land and economic hinterland development in the lower Yellow River region to a certain extent and promote urban health and sustainability in the process of new urbanization in the lower Yellow River region. Development is of great significance. The results can also provide reference for the study of spatial and temporal agglomeration characteristics, coordination and sustainability of urban economic hinterland and built-up land development in developing countries or regions similar to China's urban development model.

## 2. Materials and Methods

### 2.1. Study Area

Referring to the existing research results [41], this study defines the lower reaches of the Yellow River as an area of 20 prefecture-level cities in Henan and Shandong provinces (Figure 1), which covers a total land area of $15.119 \times 10^4$ km$^2$. This study area combines the areas covered by the irrigation area of the Lower Yellow River based on the close relationship between regional economic development and the lower reaches of the Yellow River as well as the integrity of prefecture-level administrative divisions. As Laiwu was classified as the Laiwu District of Jinan City by approval of the State Council of China in 2019, this study regards it as a prefecture-level city based on the 2017 administrative division. In 2017, the total population of the lower reaches of the Yellow River was 114.278 million, the Gross Domestic Product (GDP) was 6409.996 billion yuan, and the area of urban built-up land was 3093.720 km$^2$, accounting for 8.221%, 7.810%, and 5.609% of China's total population, GDP, and total area of urban built-up land, respectively. Combined GDP and area of urban built-up land in the cities of Zhengzhou and Jinan was 1634.540 billion yuan and 950.090 km$^2$ respectively, accounting for 25.500% and 30.710% of total GDP and area of urban built-up land in the lower reaches of the Yellow

River, respectively. As suggested by these numbers, these two core cities drive urban development and social and economic growth in the lower reaches of the Yellow River.

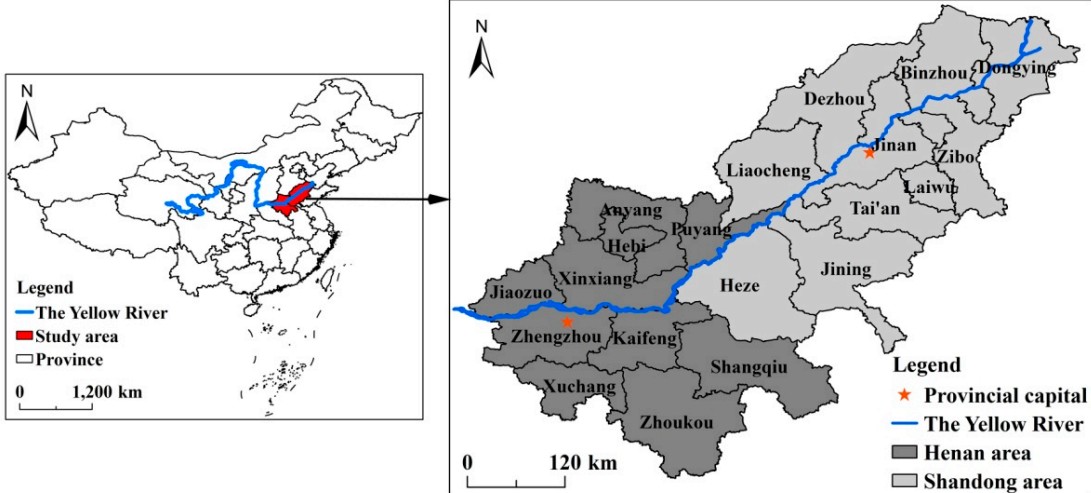

**Figure 1.** Location of the study area.

*2.2. Data*

The data used in this study mainly include road network vector data and economic and social data. Road network data came from the National Science and Technology Infrastructure of China, the Data Sharing Infrastructure of Earth System Science-Data Center of Lower Yellow River Regions (http://henu.geodata.cn), and the National Basic Geographic Database of 1:1 million (2018 edition) of China (http://www.webmap.cn); economic and social data came from the Henan Statistical Yearbook (2006–2018), the Shandong Statistical Yearbook (2006–2018), the China Urban Construction Statistical Yearbook (2005–2017), and the Portal Websites of Municipal People's Governments in the Lower Reaches of the Yellow River. The vector maps of municipal administrative divisions and the Yellow River came from the Resource and Environment Science Data Cloud Platform of the Chinese Academy of Sciences (http://www.resdc.cn).

*2.3. Methods*

2.3.1. Improved Field Model

As an effective tool to measure the intensity of regional spatial interaction, the field model has been widely used in the division of urban economic hinterland [42]. This model can reflect not only the radiation capacity of a city's economic center to the surrounding areas, but also the surrounding areas' acceptance degree of the radiation capacity of the economic center [43]. The improved field model measures the economic impact of the city by the size of and distance from the city, and is measured by the following equation [6,17,44]:

$$E_{ij} = \frac{Z_i}{D_{ij}^a} \tag{1}$$

where $E_{ij}$ represents the field strength of $i$ city to $j$ point in the region; $Z_i$ represents the economic scale of $i$ city; $D_{ij}$ represents the distance from $i$ city to $j$ point, with the shortest time distance being used in this study; and $a$ is the friction coefficient, with the standard value of 2 [8,17,20].

The centrality index is a commonly used indicator for assessing the scale of a city's economy and can be measured by the geometric mean method [45]. Because the lower reaches of the Yellow River include some cities belonging to Henan and Shandong provinces, there are some difficulties in data

convergence. Thus, the urban population and output value of secondary and tertiary industries are used to calculate the city's central index. The central index is calculated as follows [46]:

$$Z_i = \sqrt{P_i \times G_i} \tag{2}$$

where $P_i$ represents the urban population of $i$ city and $G_i$ represents the output value of secondary and tertiary industries of $i$ city.

To calculate the shortest time distance, this study uses the grid cost distance calculation method. According to the Highway Engineering Technology Standard of the People's Republic of China (JTGB01-2003), and considering the density and quality of the road network in the study area, the average driving speed of highways, national roads, provincial roads, and county and township roads is 100 km/h, 80 km/h, 60 km/h, and 30 km/h, respectively. Considering that railway traffic is more restrictive in terms of time and operating shifts and that major roads run along the railway in China [18], only the influence of road traffic is considered. At the same time, this study mainly measures the range of urban economic hinterland by the shortest time distance of road traffic, while barrier factors such as land, mountain, and water and the roads below the county and township level have a relatively small impact on the economic hinterland of the city. Therefore, the barrier factors and the roads below the county and township level are replaced by a walking speed of 6 km/h. Based on ArcGIS 10.2 software, the lower reaches of the Yellow River are divided into 150,648 effective grid elements with a size of 1 km * 1 km by using the grid analysis method. The specific technical operation process is as follows: the first step is to generate time-cost grids based on road network vector data and road speed at all levels; in the second step, the time-cost distance of each network unit in the lower reaches of the Yellow River to the central point of each municipal administrative division is obtained by using the cost distance tool in turn. In the third step, the grid calculator tool and Equation (1) are used to obtain the field intensity maps of the network units from the cities to the lower reaches of the Yellow River by combining the city center index and the time-cost distance calculated by Equation (2). The fourth step is to use the highest position tool to determine each city, from a total of 20 cities, corresponding to each network unit according to the principle of large value, so as to finally determine the economic hinterland of each city.

### 2.3.2. Geographic Concentration

The geographic concentration degree is an important index to measure the spatial concentration degree of a certain element in a region [47]. Referring to the calculation method of population and economic geographic concentration [48,49], models of the geographic concentration degree of built-up land and economic hinterland, respectively, are established to quantify their spatial concentration degree. The calculation equations are as follows:

$$R_{CLi} = \frac{CLi \sum CLi}{s_i \sum s_i} \tag{3}$$

$$R_{ERi} = \frac{ERi \sum ERi}{s_i \sum s_i} \tag{4}$$

where $R_{CLi}$ and $R_{ERi}$ indicate the geographic concentration of built-up land and economic hinterland of $i$ city, respectively; $CLi$ and $ERi$ represent the built-up land area and economic hinterland area of $i$ city, respectively; $Si$ represents the territorial area of $i$ city; and $\sum$ is the cumulative calculation of certain attributes in the entire research area.

### 2.3.3. Elasticity Coefficient of Built-Up Land and Economic Hinterland

The elasticity coefficient refers to the ratio of the change rate of two variables, which can quantitatively identify the degree of coupling development between two variables [50]. Based on the elasticity coefficient model of urban expansion [51], this study establishes the elasticity coefficient model

of the development of built-up land and economic hinterland, which reflects the coupling relationship between the development of urban built-up land and urban economic hinterland. The elasticity coefficient is calculated as follows:

$$EC_{im} = \frac{CL_{im}}{ER_{im}} = \frac{(CL_{ij} - CL_{i0})/CL_{i0}}{(ER_{ij} - ER_{i0})/ER_{i0}} \tag{5}$$

where $EC_{im}$ is the elasticity coefficient of the development of urban built-up land and urban economic hinterland in $m$ period of $i$ city; $CL_{im}$ and $ER_{im}$ represent the change rate of built-up land area and the change rate of economic hinterland in $m$ period of $i$ city, respectively; $CL_{ij}$ and $ER_{ij}$ represent the built-up land area and economic hinterland in $j$ year of $i$ city, respectively; and $CLi0$ and $ERi0$ represent the built-up land area and economic hinterland in $i$ city at the base period, respectively. The model can reflect the relative equilibrium level between the expansion of urban built-up land area and the change in urban economic hinterland. Because $CL_{im}$ is always positive in the research period, this study classifies the coupling types of urban built-up land and economic hinterland into three types—growth, intensive, and extensive—by analyzing the changes in $ER_{im}$ and $EC_{im}$ and drawing lessons from the classification methods of Long [52]. The characteristics of the coupling relationship between them are then summarized (Table 1).

**Table 1.** Classification and characteristics of elasticity coefficient of urban built-up land and economic hinterland. ($ER_{im}$, change rate of economic hinterland in $m$ period of $i$ city; $EC_{im}$, the elasticity coefficient of the development of urban built-up land and urban economic hinterland in $m$ period of $i$ city.)

| Type | $ER_{im}$ | $EC_{im}$ | Characteristic |
|------|-----------|-----------|----------------|
| growth type | $ER_{im} > 0$ | $EC_{im} > 1$ | The growth rate of built-up land is faster than that of economic hinterland |
| intensive type | $ER_{im} > 0$ | $0 < EC_{im} < 1$ | The growth rate of built-up land is slower than that of economic hinterland |
| extensive type | $ER_{im} < 0$ | $EC_{im} < 0$ | Reduction of economic hinterland and increase in built-up land |

## 3. Results and Discussion

### 3.1. Evolution of Spatial and Temporal Patterns in Urban Economic Hinterland

Using ArcGIS 10.2 software and based on Equations (1) and (2), the economic hinterland of each city in the lower reaches of the Yellow River in 2005–2017 was divided, and the division results were superimposed with administrative boundaries to obtain the spatial distribution pattern of economic hinterland in each city (Figure 2). Further sorting of the economic hinterland of each city was conducted, and the proportion of the hinterland area to the entire area is shown in Table 2.

From the spatial perspective, the pattern of economic hinterland of each city in the lower reaches of the Yellow River from 2005 to 2017 was basically the same, showing low consistency with administrative divisions and obvious differences in urban economic hinterland areas (see Figure 2). The economic hinterland of Zhengzhou and Jinan covers most of the lower reaches of the Yellow River, indicating that they have obvious squeezing effects on their surrounding cities. The two cities jointly determine the distribution pattern of the dual core of the economic hinterland in the lower Yellow River: in Henan Province, Kaifeng, Xinxiang, Xuchang, Jiaozuo, Zhoukou, and other cities suffered from the "erosion" phenomenon in Zhengzhou; in Shandong Province, Jinan has "encroached" upon Dezhou, Binzhou, Tai'an, Liaocheng, and Laiwu. These cities are near Zhengzhou and Jinan, and their own economic hinterlands are not sufficient to cover the scope of their administrative divisions. This indicates that the core cities will inhibit the development of economic hinterlands in surrounding cities. Anyang, Puyang, Hebi, Shangqiu, and other cities in Henan, as well as Jining, Heze, Zibo, and Dongying in Shandong are relatively far away from the two core cities, and thus the squeezing effect was small.

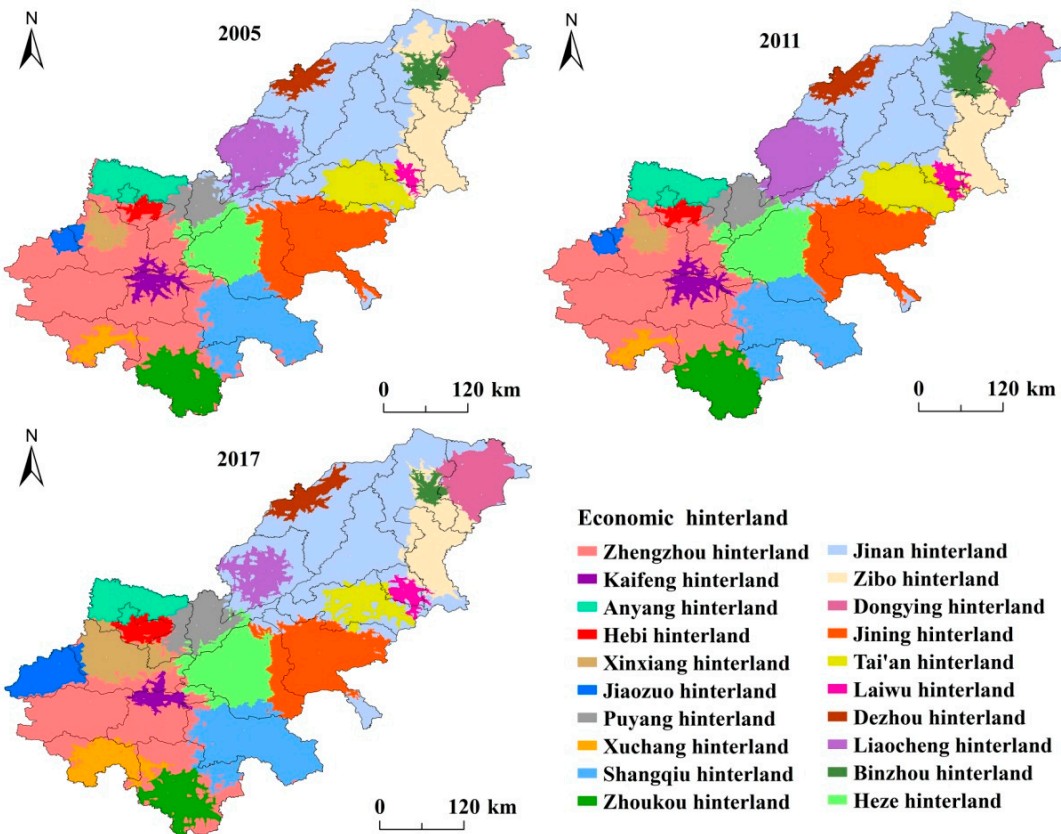

**Figure 2.** Spatial pattern changes in urban economic hinterland during the period 2005–2017.

From a time perspective, the differences in the range of economic hinterland in the lower reaches of the Yellow River were more significant (see Table 2). In terms of quantity, Zhengzhou had the largest amount of economic hinterland in 2005 and 2011, with 21.281% and 19.679% of the hinterland of the study area, respectively. Jinan followed closely, with 18.971% of the total hinterland in 2005 and 18.06% in 2011. In 2017, Jinan surpassed Zhengzhou and held the largest area of hinterland—23.842%—which was 1.631 times that of Zhengzhou. The cities with the smallest economic hinterland in 2005–2017 were Laiwu, Jiaozuo, and Binzhou, with 0.484%, 0.692% and 0.739%, respectively. Over time, the quantitative gap between the largest economic hinterland and the smallest economic hinterland has narrowed. In terms of change, 12 cities increased their share of hinterland to varying degrees, and Zhengzhou, Zibo, Liaocheng, Jining, Tai'an, Zhoukou, Kaifeng, and Dongying saw reductions of economic hinterland. Specifically, the top three cities in terms of reduction of economic hinterland were Zhengzhou, Zibo, and Liaocheng—all of which were reduced by more than 2723 km$^2$, with the largest reduction in Zhengzhou, by 10,032 km$^2$. The top three cities in terms of increases were Jinan, Xinxiang, and Jiaozuo—all of which were increased by more than 2930 km$^2$, and with the largest increase in Jinan by 7338 km$^2$. Through the above analysis, we find that the gap between the urban economic hinterland in Shandong Province is expanding, and the gap between the urban economic hinterland in Henan Province is decreasing. The main reason is that the economic hinterland of Jinan, the capital city of Shandong Province, is more competitive with its surrounding cities, such as Zibo and Liaocheng, which has resulted in a large increase in the economic hinterland of Jinan. The economic hinterland of Zhengzhou, the capital city of Henan Province, has less radiation effect on its surrounding cities, such as Xinxiang and Jiaozuo, which has resulted in the large-scale reduction of the economic hinterland of Zhengzhou, thus reducing the gap between Zhengzhou and other cities in Henan Province.

**Table 2.** The area and proportion of economic hinterland during 2005–2017.

| City | 2005 Year | | 2011 Year | | 2017 Year | |
|---|---|---|---|---|---|---|
| | Area/km$^2$ | Proportion/% | Area/km$^2$ | Proportion/% | Area/km$^2$ | Proportion/% |
| Zhengzhou | 32,060 | 21.281 | 29646 | 19.679 | 22028 | 14.622 |
| Kaifeng | 2242 | 1.488 | 2613 | 1.735 | 1650 | 1.095 |
| An'yang | 4445 | 2.951 | 4864 | 3.229 | 4717 | 3.131 |
| Hebi | 976 | 0.648 | 1103 | 0.732 | 1898 | 1.260 |
| Xinxiang | 2041 | 1.355 | 2102 | 1.395 | 6114 | 4.058 |
| Jiaozuo | 1150 | 0.763 | 1042 | 0.692 | 4080 | 2.708 |
| Puyang | 3339 | 2.216 | 3900 | 2.589 | 4607 | 3.058 |
| Xuchang | 1831 | 1.215 | 1539 | 1.022 | 4327 | 2.872 |
| Shangqiu | 12,056 | 8.003 | 13350 | 8.862 | 12336 | 8.189 |
| Zhoukou | 6153 | 4.084 | 6930 | 4.600 | 4647 | 3.085 |
| Jinan | 28,580 | 18.971 | 27126 | 18.006 | 35918 | 23.842 |
| Zibo | 11,295 | 7.498 | 7140 | 4.740 | 7841 | 5.205 |
| Dongying | 6034 | 4.005 | 6210 | 4.122 | 6020 | 3.996 |
| Jining | 14,503 | 9.627 | 13346 | 8.859 | 12054 | 8.001 |
| Tai'an | 5507 | 3.656 | 5768 | 3.829 | 3892 | 2.584 |
| Laiwu | 729 | 0.484 | 1196 | 0.794 | 1319 | 0.876 |
| Dezhou | 1875 | 1.245 | 2305 | 1.530 | 2503 | 1.661 |
| Liaocheng | 6522 | 4.329 | 7866 | 5.221 | 3799 | 2.522 |
| Binzhou | 1726 | 1.146 | 3551 | 2.357 | 1113 | 0.739 |
| Heze | 7584 | 5.034 | 9051 | 6.008 | 9785 | 6.495 |

*3.2. Analysis of Spatial and Temporal Agglomeration Characteristics of Built-Up Land and Economic Hinterland*

The geographic concentration of built-up land and economic hinterland of each city in the lower reaches of the Yellow River were measured using Equations (3) and (4). To facilitate a comparison, the calculation results were classified and the spatial distribution map of geographic concentration (Figures 3 and 4) was drawn, which more clearly illustrates the evolution characteristics and trends of urban built-up land and economic hinterland in the lower Yellow River.

As shown in Figure 3, the geographic concentration of built-up land in the lower reaches of the Yellow River from 2005 to 2017 centers on Zhengzhou and Jinan and spreads to the surrounding areas, gradually forming high-value contiguous zones around Zhengzhou and Jinan. In 2005, the geographic concentration of built-up land in the lower reaches of the Yellow River was generally low. At this time, the spatial distribution of built-up land was relatively balanced. In 2005, the value of built-up land was 0.025–0.050, with the exception of Zhengzhou, Jinan, Zibo, and Laiwu, which were all below 0.025. By 2011, although the geographic concentration of built-up land had increased, the increase was not significant: Zhengzhou, Jinan, and Zibo entered the range of 0.051–0.100, Hebi and Jiaozuo increased from 0.018 to 0.040 and 0.035 respectively, while the rest of the cities remained basically unchanged. By 2017, the imbalance in spatial distribution of built-up land became evident, basically forming a double "core circle" structure with Zhengzhou and Jinan as the core and spreading to the periphery. During the study period, the total area of built-up land in the lower reaches of the Yellow River increased from 1559.080 km$^2$ in 2005 to 3093.720 km$^2$ in 2017, nearly doubling in 12 years, while the total geographic concentration increased from 0.251 to 0.929, increasing by 3.701 times. This further demonstrates that the total amount of built-up land increased while the spatial distribution became more concentrated. During the study period, the total geographic concentration of built-up land in the lower reaches of the Yellow River increased from 0.251 in 2005 to 0.929 in 2017 and increased by 2.701 times in 12 years. The main reason is that most cities in the lower reaches of the Yellow River show multi-center expansion in the process of development. Population and capital around the city tend to concentrate in the urban center with a larger economic total [53]. As a result, the relationship between people and land in the city becomes increasingly tense and land for urban construction begins

to increase. From 2005 to 2017, the total area of built-up land in the lower reaches of the Yellow River increased from 1559.080 km$^2$ to 3093.720 km$^2$. The continuous expansion of built-up land area led to the increasing geographic concentration of construction land.

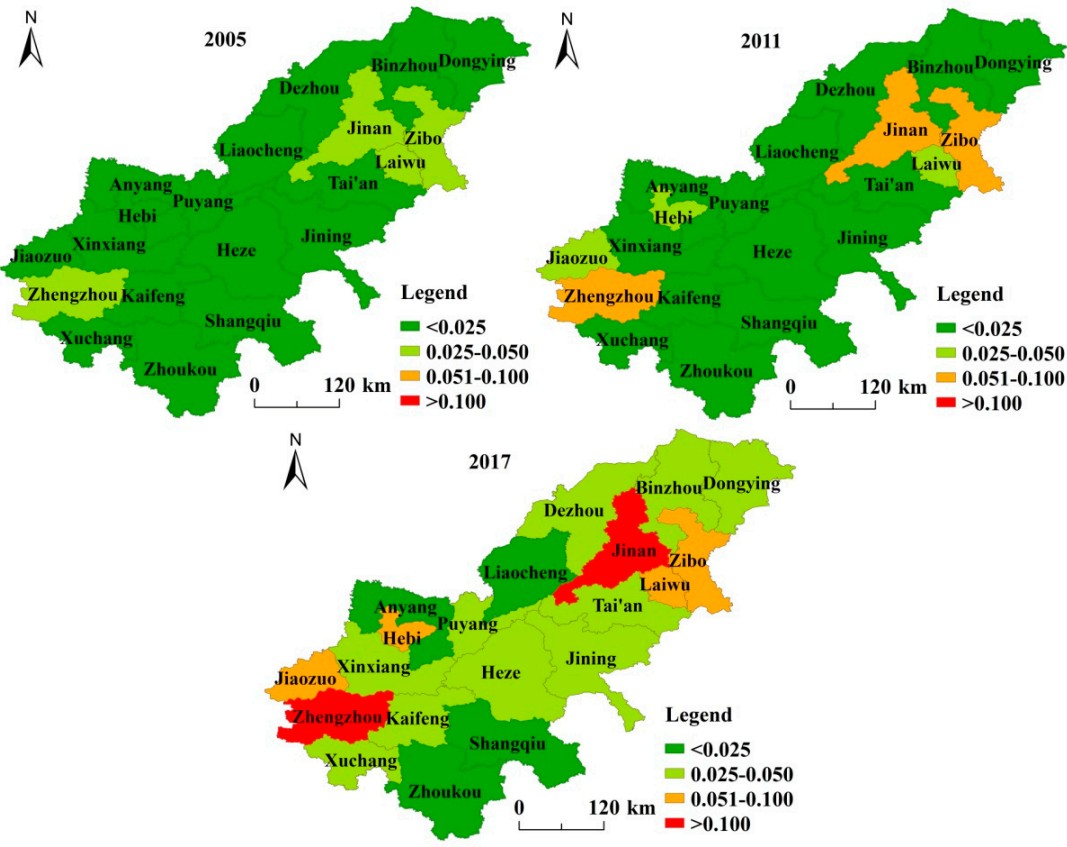

**Figure 3.** Distribution of urban built-up land concentration during the period 2005–2017.

Figure 4 shows that the spatial distribution pattern of urban economic hinterland in the lower reaches of the Yellow River in 2005–2017 changed less than that of built-up land. The economic hinterland was centered on Zhengzhou and Jinan, with a "highest-low-higher" value circle structure, and this pattern gradually weakened in Henan and was slightly strengthened in Shandong. At the three research time nodes (2005, 2011, and 2017), the geographic concentration of Zhengzhou and Jinan was over 2.940 and occupied the highest value area. The low value area was generally near the surrounding cities of Zhengzhou and Jinan, and the changing cities mainly develop from the low value to 0.500–1.000. From 2005 to 2011, the change was small, which demonstrates that Hebi in Henan Province was less affected by radiation from Zhengzhou and that Laiwu in Shandong Province was less affected by radiation from Jinan. The geographic concentration of the economic hinterland of the two cities increased from 0.446 and 0.323 in 2005 to 0.504 and 0.531 in 2011, respectively. By 2017, the values of Xuchang, Jiaozuo, and Xinxiang entered the range of 1.001–2.000, which was an increase from 0.500–1.000 in 2011; as a result, the "highest-low-higher" circle structure of geographic concentration in the economic hinterland of Henan Province was broken. At the same time, Puyang increased from 0.928 in 2011 to 1.096 in 2017, which reflected a one-level increase, and Zhoukou decreased from 0.577 to 0.387, which reflected a one-level decrease. However, Tai'an and Liaocheng dropped from the 0.500–1.000 range to the low value area, which formed a more obvious "highest-low-higher" spatial structure in Shandong Province. Therefore, the equilibrium of the development trend of Henan urban economic hinterland in the lower reaches of the Yellow River in the study period was lower than that of Shandong.

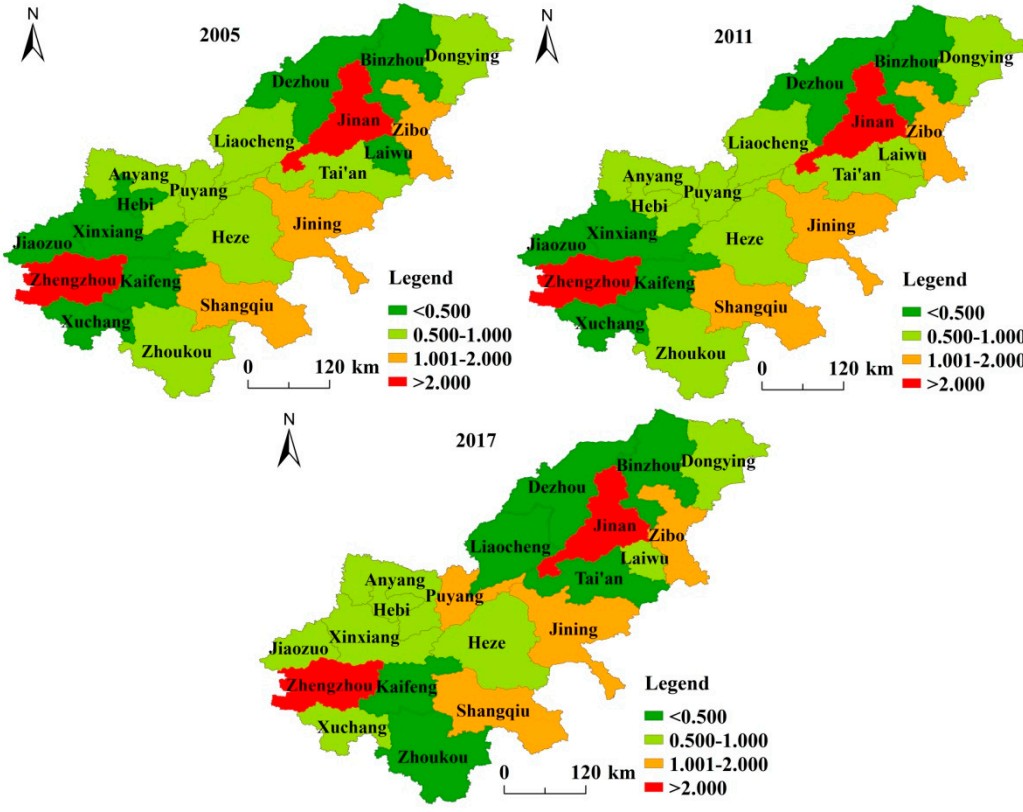

**Figure 4.** Distribution of urban economic hinterland concentration during the period 2005–2017.

Based on the geographic concentration index of urban built-up land and economic hinterland in the lower reaches of the Yellow River, the fit curve was established by drawing scatter plots (Figure 5). The results show that there was no significant correlation between the geographic concentration of built-up land and economic hinterland in 2005 (the goodness of fit was 0.482). However, with the expansion of urban built-up land, there was a moderate positive correlation in 2011–2017 (the goodness of fit was 0.793 and 0.743), indicating that the spatial agglomeration of built-up land and economic hinterland at the city scale in the lower reaches of the Yellow River had a certain correlation—that is, the larger the area of built-up land, the larger the economic hinterland. However, the degree of correlation was not high, and there were still some cities where the expansion of built-up land did not match the development of their own economic hinterland.

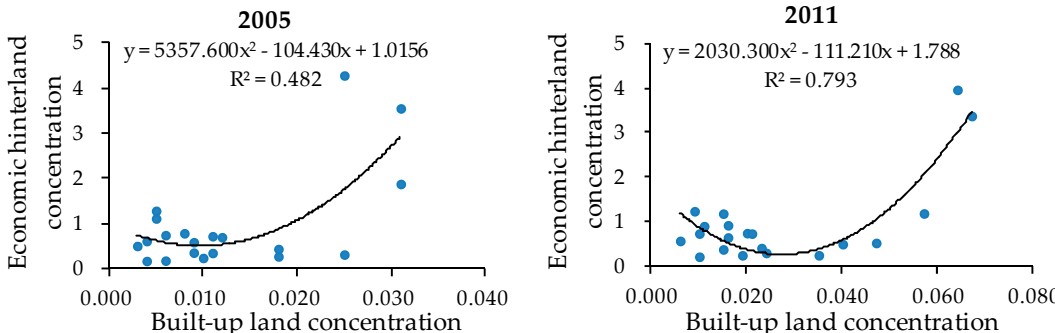

**Figure 5.** *Cont.*

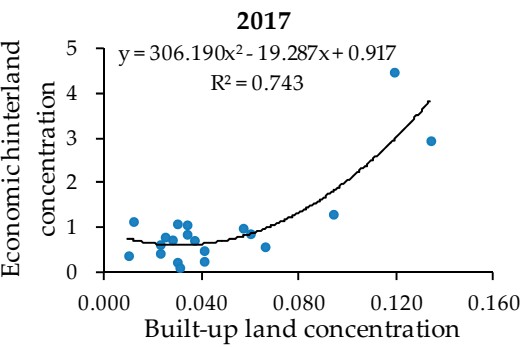

**Figure 5.** Fit curve of urban built-up land and economic hinterland in 2011 and 2017.

### 3.3. Spatial–Temporal Coupling Relationship between Built-Up Land Expansion and Economic Hinterland Development

Using Equation (5) and Table 1, the elasticity coefficient of urban built-up land and economic hinterland in the lower reaches of the Yellow River from 2005 to 2017 was calculated (Figure 6). The spatial–temporal coupling characteristics of urban built-up land and economic hinterland development were further analyzed (Figure 7).

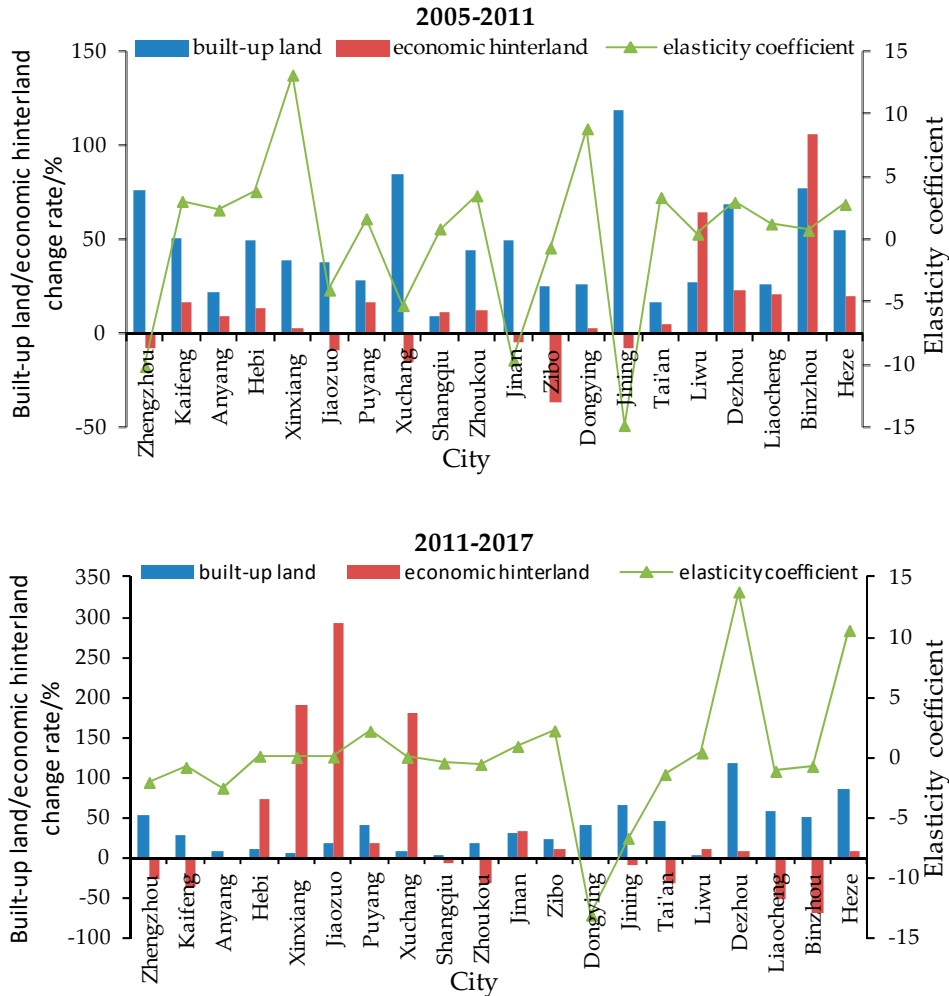

**Figure 6.** Urban built-up land and economic hinterland change rate and the elasticity coefficient during the period 2005–2017.

As shown in Figure 6, the area of built-up land in the lower reaches of the Yellow River continued to increase from 2005 to 2017, and the growth and decline of economic hinterland coexisted. The rate of built-up land and economic hinterland and the elasticity coefficient of built-up land and economic hinterland varied greatly between cities. From 2005 to 2011, the change rate of economic hinterland in Binzhou, Laiwu, Shangqiu, and Zibo was greater than that of built-up land, while the change rate of built-up land of other cities was greater than that of economic hinterland. In terms of the change in built-up land, Jining had the largest change rate of built-up land (up to 118.406%), whereas Shangqiu had the smallest change rate of built-up land (9.145%), and other cities were between 15.887% and 84.161%. In terms of the change in economic hinterland, Zibo, Xuchang, Jiaozuo, Jining, Zhengzhou, and Jinan decreased from 2005 to 2017, while the other cities increased. Of the cities that experienced an increase in the economic hinterland change rate, Binzhou had the largest (105.736%) and Dongying had the smallest (2.917%). In terms of the elasticity coefficient, Xinxiang had the highest value of 13.126, while Jining had the lowest value of −14.842. At the same time, the elasticity coefficient of Shandong region tended to be slightly more balanced than that of Henan region. From 2011 to 2017, the regional characteristics of the change in built-up land and economic hinterland and their elasticity coefficients were evident, which were mainly manifested in the increase in the change rate of economic hinterland in Henan and the increase in the change rate of built-up land in Shandong. Among them, Jiaozuo replaced Binzhou as the city with the largest change rate of economic hinterland, and Dezhou replaced Jining as the city with the largest change rate of built-up land. During this period, the elasticity coefficient of built-up land and economic hinterland in Henan was more stable than that in Shandong.

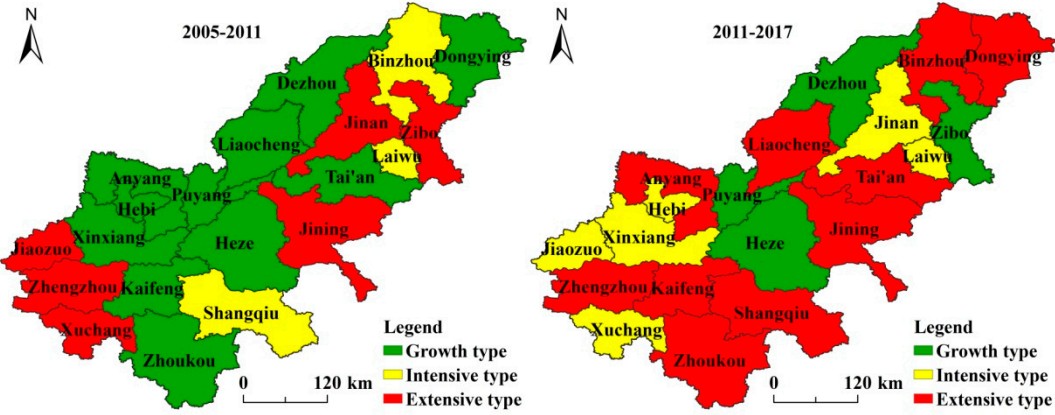

**Figure 7.** Spatial–temporal patterns of development types of urban built-up land and economic hinterland during the period 2005–2017.

As shown in Figure 7, the coupling model of built-up land and economic hinterland in the lower reaches of the Yellow River from 2005 to 2011 was mainly growth—that is, the built-up land and economic hinterland of most cities were growing rapidly and the elasticity coefficient was greater than 1. Extensive development accounted for 30% of the cities, including Zhengzhou, Xuchang, Jinan, and Jining, while intensive development accounted for 15% of the cities, and consisted of only Shangqiu, Laiwu and Binzhou. Most cities were in growth development (55%), including Kaifeng, Zhoukou, Taian, and Liaocheng. The results reflected a non-intensive development trend in the urban built-up land and economic hinterland in the lower reaches of the Yellow River from 2005 to 2011 and that the growth rate of built-up land was too fast, which was not in harmony with the development of economic hinterland. From 2011 to 2017, the coupling model of urban built-up land and economic hinterland began to develop towards extensive and intensive development. Extensive development accounted for 50% of the cities—Zhengzhou, Kaifeng, Jining, Taian, and 10 other cities—and intensive development accounted for 30% of the cities, including Xuchang, Jiaozuo, Jinan, and Laiwu, and six other cities. Only four cities—Puyang, Zibo, Dezhou, and Heze—were considered growth development, which accounted for 20% of the cities. By comparing the changes in the coupling relationship between

urban built-up land and economic hinterland development in the two periods of 2005–2011 and 2011–2017, the development process of an economic society and new urbanization in the lower reaches of the Yellow River have a strong trend, but the overall urban built-up land expansion appears too fast.

## 4. Conclusions

By calculating the economic hinterland of the cities in the lower reaches of the Yellow River using the improved field model, the spatial agglomeration characteristics of urban built-up land and the economic hinterland were analyzed by considering geographic concentration and the elasticity coefficient. The following conclusions were drawn:

(1) From 2005 to 2017, the spatial and temporal patterns of the economic hinterland in the lower reaches of the Yellow River were basically the same, showing the characteristics of low coincidence with administrative divisions and obvious differences in the scope and changes in the economic hinterland. The economic hinterland of Zhengzhou, Zibo, and Liaocheng decreased the most, while Jinan, Xinxiang, and Jiaozuo increased the most. The economic hinterland of Zhengzhou and Jinan always covered most of the lower reaches of the Yellow River; the economic hinterland of Zhengzhou was the largest in 2005–2011 and that of Jinan was the largest in 2017.

(2) From 2005 to 2017, the geographic concentration of built-up land in the lower reaches of the Yellow River showed a trend of centering on Zhengzhou and Jinan and spreading to the surrounding areas, gradually forming high-value contiguous zones around both Zhengzhou and Jinan. Compared with the built-up land, the spatial distribution pattern of geographic concentration in the economic hinterland had a smaller change. The overall pattern reflected Zhengzhou and Jinan as the core, showing a "highest-low-higher" circle structure moving away from the core, and this pattern gradually weakened in Henan and slightly strengthened in Shandong. With the development of cities, the spatial agglomeration of urban built-up land and economic hinterland in the lower reaches of the Yellow River began to match the development trend. The geographic concentration of built-up land and hinterland in Zhengzhou and Jinan always maintained the highest value in the region.

(3) The spatial–temporal coupling relationship between urban built-up land expansion and economic hinterland development in the lower reaches of the Yellow River was obvious. In 2005–2011, the coupling model of the development of built-up land and economic hinterland in the lower reaches of the Yellow River was mainly growth, with growth, extensive, and intensive development cities accounting for 55%, 30%, and 15% of the cities in the study area, respectively. In 2011–2017, the coupling model began to shift towards extensive and intensive development, with extensive, intensive, and growth development cities accounting for 50%, 30%, and 20% of the cities in the study area, respectively. Although the coupling relationship reflects a strong trend, the rapid expansion of urban built-up land is quite problematic.

A coordinated and intensive urban development system is a basic requirement of new urbanization. The extensive urban development trend is obviously contrary to the concept of new urbanization development. Therefore, in the urban development process, we should gradually challenge the tendency to drive urban economic growth through the expansion of built-up land. Cities should unblock and broaden the channels of circulation of various economic production factors in the region, increase the development of high-tech industries, constantly cultivate new economic growth points, accelerate the formation of a new pattern of economic development to promote the intensive use of urban land, and establish a benign interaction between built-up land and economic hinterland.

The objective of this study was to examine the spatial–temporal coupling relationship between urban built-up land and the development of economic hinterland. However, in the process of new urbanization, there may be many factors closely related to the development of urban impact areas, such as population, urban cultural influences, and urban ecological environment. Thus, it may be valuable to determine the relationship between these factors and the development of urban impact areas in future studies to promote urban sustainable development.

**Author Contributions:** All authors conceived, designed, and implemented the study. P.Z. and Y.C. designed and carried out the study. W.J., Y.Y. and Y.Z. participated in the analysis and presentation of analytic results. Y.L. and D.Y. collected and analyzed data. X.L., W.G. and T.R. contributed the data used in this study.

**Funding:** This research was funded by the National Natural Science Foundation of China, grant number 41601175, 41801362, the key scientific research project of Henan province, grant number 16A610001, the 2018 Young Backbone Teachers Foundation from Henan Province, grant number 2018GGJS019, the Program for Innovative Research Talent in University of Henan Province, grant number 20HASTIT017, Key R&D and extension projects in Henan Province in 2019 (agriculture and social development field), grant number 192102310002, the Innovation Team Cultivation Project of The First-class Discipline in Henan University, grant number 2018YLTD16, the Natural Science Foundation of Guangdong Province, China, grant number 2018A030310470, the GDA'S Project of Science and Technology Development, grant number 2016GDASRC-0211, 2017GDASCX-0601, 2018GDASCX-0403, 2019GDASYL-0502001, the Guangdong Innovative and Entrepreneurial Research Team Program, grant number 2016ZT06D336.

**Acknowledgments:** This study was jointly supported by the National Natural Science Foundation of China (41601175, 41801362), the key scientific research project of Henan province (16A610001), the 2018 Young Backbone Teachers Foundation from Henan Province (2018GGJS019), the Program for Innovative Research Talent in University of Henan Province (20HASTIT017), Key R&D and extension projects in Henan Province in 2019 (agriculture and social development field) (192102310002), the Innovation Team Cultivation Project of The First-class Discipline in Henan University (2018YLTD16), the Natural Science Foundation of Guangdong Province, China (2018A030310470), the GDAS Project of Science and Technology Development (2017GDASCX-0101, 2018GDASCX-0101, 2019GDASYL-0302001), the Guangdong Innovative and Entrepreneurial Research Team Program (2016ZT06D336), the Forest Science and Technology Innovation in Guangdong (2015KJCX047), and the National Earth System Science Data Sharing Infrastructure (http://www.geodata.cn/).

**Conflicts of Interest:** All authors declare no conflict of interest.

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
