# Peer review of "Spatial and Temporal Agglomeration Characteristics and Coupling Relationship of Urban Built-Up Land and Economic Hinterland—A Case Study of the Lower Yellow River, China"

_sustainability, doi:10.3390/su11195218_

Round 1

Reviewer 1 Report

There may be the kernel of a publishable paper here, but it is hard to find. For the vast majority of the Sustainability audience who isn’t familiar with the geographic context, much more is needed to describe the focal region, whether it is representative of other regions in China, and how it compares to more global regions. The analysis has a few interesting features, but is remarkably basic in approach. Its major pillars are a simple gravity model, without empirically evaluating the standard squared term (p4), an even simpler elasticity coefficient (p5), and a regression analysis of a tiny sample which is clearly driven by outliers (Figure 5 on p10). Figure 6 is completely opaque in terms of its implications, not helped by a seemingly random ordering of the regional outcomes.

Author Response

Comment 1: There may be the kernel of a publishable paper here, but it is hard to find.

Response: Thank you for your comments. After consulting a large number of articles about urban hinterland, we find that most scholars only pay attention to the empirical research on the definition of urban hinterland, and do not further explore the relationship between urban hinterland and other socio-economic factors. Developing countries like China are experiencing rapid economic and urbanization development in recent years, accompanied by the drastic expansion of urban built-up land. In this process, whether the development relationship between urban economic hinterland and built-up land is coordinated is directly related to the health and sustainable development of the city. The purpose of this paper is to empirically study the space-time development relationship between urban economic hinterland and built-up land, and it is also the innovation of this paper. To some extent, our research results can reflect the coordination of urban economic hinterland and built-up land development in the lower Yellow River, which is a typical area of urban development in China. It can provide theoretical support for the strategic deployment of urban healthy development in the lower Yellow River,it can also provide a scientific reference for the study of urban sustainable development in China and other developing countries similar to China's urban development model.

Comment 2: For the vast majority of the Sustainability audience who isn’t familiar with the geographic context, much more is needed to describe the focal region, whether it is representative of other regions in China, and how it compares to more global regions.

Response: Thank you for your comments. We have added more descriptions of research areas to the revised edition. For more information, see line 101-113 on page 3.

Comment 3: The analysis has a few interesting features, but is remarkably basic in approach. Its major pillars are a simple gravity model, without empirically evaluating the standard squared term (p4), an even simpler elasticity coefficient (p5), and a regression analysis of a tiny sample which is clearly driven by outliers (Figure 5 on p10). Figure 6 is completely opaque in terms of its implications, not helped by a seemingly random ordering of the regional outcomes.

Response: Thank you for your comments. The methods you mentioned in this article are relatively simple. We agree with you, but we believe that these simple methods can illustrate the spatial and temporal agglomeration characteristics and coupling relationship of economic hinterland and built-up land development in the process of urban development in typical areas of China, and can also be used for cities in the world and China. The data we used are official data of Chinese government departments and data provided by authoritative geographic websites, which can support our research scientifically and accurately. The left ordinate of figure 6 represents the ratio of urban built-up land and economic hinterland area change, which is shown in the histogram. The blue is the change rate of built-up land area, and the red is the change rate of economic hinterland area. Take the rate of change of Zhengzhou's economic hinterland area from 2005 to 2011 as an example. The calculation process is as follows :(Zhengzhou's economic hinterland area in 2011 - Zhengzhou's economic hinterland area in 2005) / Zhengzhou's economic hinterland area in 2005; The right ordinate of figure 6 shows the elasticity coefficient of the development of built-up land and economic hinterland, that is, the ratio of the change rate of built-up land area to the change rate of economic hinterland area, which is shown in the form of a broken line graph. The whole calculation process is based on formula (5). The purpose of figure 6 is to more intuitively reveal the change rate and elasticity coefficient of urban built-up land and economic hinterland of each city in the lower reaches of the Yellow River from 2005 to 2011 and 2011 to 2017.

Reviewer 2 Report

The paper identifies the urban economic hinterland in the lower reaches of the Yellow River in China using various methods (such as the improved field model, geographic concentration model, and elasticity coefficient model) and further studies the spatial and temporal link between urban built-up land expansion and economic hinterland development. The results suggest that appropriate coordination between urban built-up land and economic hinterland is necessary to promote sustainable development of cities in the process of new urbanization.

Recommendations:

I suggest that the authors explain more fully the necessity to explore the spatial-temporal relationship between urban built-up land and economic hinterland in the specific context of regional development in China.

Please check again the Classification in Table 1. If I am not mistaken, in the case of intensive type, if ECim> 1, the growth rate of built-up land, which is the numerator of the fraction (5), should be faster than that of economic hinterland (the denominator).

It is not clear to me how did the authors determine the economic hinterland of each city based on equations (1) and (2). How did you combine them? Moreover, I understand that the area of influence decreases with the distance from the city centre, but how did you establish the threshold, the limit beyond which the effect becomes null? Please explain in more detail.

Lines 224-237. The authors present the changes in terms of increased or decreased share of hinterland for the cities under investigation. Maybe they could add some explanations for these developments.

In the same register, what are the factors behind the increase in geographic concentration of built-up land in the lower reaches of the Yellow River from 2005 to 2017?

Author Response

Reviewer #2

Comment 1: I suggest that the authors explain more fully the necessity to explore the spatial-temporal relationship between urban built-up land and economic hinterland in the specific context of regional development in China.

Response: Thank you for your reminder. We have added the content of "the spatial-temporal relationship between urban built-up land and economic hinterland in the specific context of regional development in China" in the revised version. For more information, see line 88-99 on page 2.

Comment 2: Please check again the Classification in Table 1. If I am not mistaken, in the case of intensive type, if ECim> 1, the growth rate of built-up land, which is the numerator of the fraction (5), should be faster than that of economic hinterland (the denominator).

Response: Thank you very much for your careful review. We have corrected Table 1 in the revised edition. For more information, see Table 1 on page 6.

Comment 3: It is not clear to me how did the authors determine the economic hinterland of each city based on equations (1) and (2). How did you combine them?

Response: Thank you for your comments. We have added "determine the economic hinterland of each city based on equations (1) and (2)" to the revised version. For more information, see line 187-197 on page 5.

Comment 4: Moreover, I understand that the area of influence decreases with the distance from the city centre, but how did you establish the threshold, the limit beyond which the effect becomes null?

Response: Thank you for your comments. The impact does decrease with the distance from the city center. There are two main determinants of this threshold. One is the value of friction coefficient a in formula (1). According to the habitual practices of Chinese scholars (see references 8, 17 and 20, respectively) in studying urban hinterland in China, the standard value of a is 2. The other, in the final step of determining the economic hinterland of each city, according to the principle of taking the maximum value of the field strength of 20 cities corresponding to 150 648 1 km *1 km network unit, the highest position tool in the ArcGIS10.2 was used to determine the economic hinterland of each city, i.e. each city.  If the field strength of each network unit is the largest corresponding to a city, the network unit belongs to that city. I hope our explanation can settle your doubts. These two determinants are already covered in the revised version. Please refer to line 168 on page 5 and line 187-197 on page 5, respectively.

Comment 5: Lines 224-237. The authors present the changes in terms of increased or decreased share of hinterland for the cities under investigation. Maybe they could add some explanations for these developments.

Response: Thank you for your comments. In the revised edition, we have added the reasons why the gap between urban economic hinterlands in Shandong Province has increased and the gap between urban economic hinterlands in Henan Province has decreased. For more information, please refer to line 266-274 on page 7.

Comment 6: In the same register, what are the factors behind the increase in geographic concentration of built-up land in the lower reaches of the Yellow River from 2005 to 2017?

Response: Thank you for your comments. In the revised edition, the reasons for the increase in geographic concentration of built-up land in the lower reaches of the Yellow River from 2005 to 2017 have been appropriately added. For more information, see line 299-308 on page 9.

Reviewer 3 Report

This paper will be in good shape to be published, after the following related studies at the city/urban level will be cited in the text:

“Trade and Cities” World Bank Economic Review, 2015, 29 (3): 523-549 “Agglomeration and Trade: State-Level Evidence from U.S. Industries,” Journal of Regional Science, 2011, 51(1): 139-166. “Anti-Crime Laws and Retail Prices” Review of Law and Economics, 2017, 13(3): 1-18. “Understanding Long-run Price Dispersion", Journal of Monetary Economics, 2014, 66: 226-240. “Price-Level Convergence: New Evidence from U.S. Cities”, Economics Letters, 2011, 110: 76-78.

Author Response

Comment 1: This paper will be in good shape to be published, after the following related studies at the city/urban level will be cited in the text:

Response: Thank you very much for your comments on this article. We have carefully read the five references you have designated and agree with them very much, but unfortunately only three of them have been cited, and the other two are different from the contents of this paper. For more information, please refer to references 2, 3 and 53, respectively.

Round 2

Reviewer 1 Report

The authors have done a reasonable job in better contextualizing and motivating the paper. I now agree with my fellow reviewers that it is ready for publication in Sustainability, despite being a relatively marginal contribution to the literature.

Author Response

Thanks for your suggestions. Using the improved field model, geographic concentration and elasticity coefficient, and taking the lower reaches of the Yellow River as the study area, this study defines the scope of urban economic hinterland in the lower reaches of the Yellow River in China. It can provide theoretical support for the strategic deployment of urban healthy development in the lower Yellow River,it can also provide a scientific reference for the study of urban sustainable development in China and other developing countries similar to China's urban development model. We will consider your opinions carefully and make further improvements in future research. Thanks again for your comments.